# Expression and Signaling of β-Adrenoceptor Subtypes in the Diabetic Heart

**DOI:** 10.3390/cells9122548

**Published:** 2020-11-26

**Authors:** Betul R. Erdogan, Martin C. Michel, Ebru Arioglu-Inan

**Affiliations:** 1Department of Pharmacology, Faculty of Pharmacy, Ankara University, 06560 Ankara, Turkey; brerdogan@ankara.edu.tr; 2Department of Pharmacology, Faculty of Pharmacy, Izmir Katip Celebi University, 35620 Izmir, Turkey; 3Department of Pharmacology, Johannes Gutenberg University, 55131 Mainz, Germany; marmiche@uni-mainz.de

**Keywords:** diabetes, heart, beta adrenoceptor

## Abstract

Diabetes is a chronic, endocrine disorder that effects millions of people worldwide. Cardiovascular complications are the major cause of diabetes-related morbidity and mortality. Cardiac β_1_- and β_2_-adrenoceptor (AR) stimulation mediates positive inotropy and chronotropy, whereas β_3_-AR mediates negative inotropic effect. Changes in β-AR responsiveness are thought to be an important factor that contributes to the diabetic cardiac dysfunction. Diabetes related changes in β-AR expression, signaling, and β-AR mediated cardiac function have been studied by several investigators for many years. In the present review, we have screened PubMed database to obtain relevant articles on this topic. Our search has ended up with wide range of different findings about the effect of diabetes on β-AR mediated changes both in molecular and functional level. Considering these inconsistent findings, the effect of diabetes on cardiac β-AR still remains to be clarified.

## 1. Introduction

Diabetes is an endocrine disorder due to partial or complete insulin deficiency or to insulin resistance in the target tissues. According to the World Health Organization (WHO) the number of diabetic individuals rose from 108 million in 1980 to 422 million in 2014 [1]. The number of people with diabetes is estimated to rise to 629 million (at ages 20–79) by 2045 [2].

Impaired glucose uptake to tissues in the absence of an insulin effect results in hyperglycemia. Despite the different characteristics and treatment strategies, both type 1 (T1DM) and type 2 diabetes (T2DM) have a similarly serious impact on the whole body because of hyperglycemia. Uncontrolled diabetes causes partly irreversible changes to various organs, which in turn lead to diabetic complications. In fact, people with T2DM are at higher risk for diabetic complications, since the diagnosis is often made when the complications already occurred [1].

Cardiovascular complications of diabetes are one of the important causes of diabetic morbidity and mortality. Diabetes is an independent risk factor for both ischemic and hemorrhagic stroke [3]. The risk of having myocardial infarction (MI) among diabetic patients without previous MI history has been found as high as nondiabetics with a previous MI history [4]. Furthermore, the prevalence of heart failure has been reported as four time higher among diabetic patients as compared to general population [5]. Although hyperglycemia is an important contributor to diabetic cardiovascular complications, regulating blood glucose is not enough to prevent them. Some clinical trials [6,7,8] have shown that major cardiovascular events were not fully prevented despite of a tight glycemic control.

β-adrenoceptors (β-AR) include the β_1_-, β_2_- and β_3_-AR subtypes [9]. β_1_- and β_2_-AR have been considered as the only subtypes until the third one was cloned in 1989 by Emorine et al. [10]. The β_3_-AR was first detected in rodent the adipose tissue where it mediates lipolysis and thermogenesis [11]. In 1996, Gauthier et al. [12] reported the presence of this subtype in human endomyocardial biopsies. Despite the clinical trials on β_3_-AR as a therapeutic target in heart failure, there is an ongoing debate on their presence in the healthy human heart [13].

Cardiac β_1_- and β_2_-AR essentially contribute to the control of inotropy and chronotropy [14,15,16]. Both subtypes are coupled to a stimulatory G protein (G_s_) [17] and their signaling pathways include stimulation of adenylyl cyclase (AC), cyclic AMP (cAMP) formation and protein kinase A (PKA) activation. Following activation of AC and cAMP production, PKA is activated and phosphorylates L type Ca^++^ channels [18]. Increased Ca^++^ influx stimulates Ca^++^ release from the sarcoplasmic reticulum (SR) and cytosolic Ca^++^ levels are elevated. This enables cardiac contraction. β_2_-AR, on the other hand, have a dual coupling. This subtype has been suggested to couple also to an inhibitory G protein (G_i_) in rat cardiomyocytes since pretreatment with pertussis toxin (PTX) resulted in an enhanced positive inotropic response [19]. Similarly, the β_2_-AR agonist-mediated contractile response or [Ca^++^]_i_ transient amplitude were increased when murine cardiomyocytes were treated with PTX [20]. Furthermore, β_2_-AR mediated inotropic effect through G_i_ coupling also involves the cAMP-PKA signaling pathway as the response was abolished in the presence of a cAMP inhibitor [20]. It has been recently reported that the β_2_-AR-G protein interaction (coupling either to G_s_ or G_i_) is regulated by local membrane charge [21].

The β_3_-AR has some structural differences as compared to β_1_- and β_2_-AR such as being less sensitive to agonist stimulated desensitization because of lack of a phosphorylation site for PKA and β-AR kinases [22,23,24]. The β_3_-AR gene of rats or mice has 79% homology with the human ortholog [25]. Hence, an interspecies difference of the expression and function of β_3_-AR [26] should be also considered. β_3_-AR are coupled to G_s_ and G_i_, and the latter mediates a negative inotropic effect in the heart through a signaling pathway including nitric oxide synthase (NOS)-nitric oxide (NO)-cyclic guanosine monophosphate (cGMP)-protein kinase G (PKG) [27,28]. The expression of β_3_-AR in the healthy human heart is limited [13]. Of note, β_3_-AR have been found upregulated in some cardiac pathologies such as heart failure [29] and other hypoxic conditions [30]. This effect has been suggested as a preventive mechanism as the heart is exposed to overstimulation by catecholamines in these pathologies [31,32]. This idea inspired clinical trials that investigate the effectiveness of a β_3_-AR agonist mirabegron on heart disease [33,34]. 

Diabetes has been shown to affect both the expression of β-AR subtypes and β-AR mediated responsiveness [35]. The changes in the expression of β-AR subtypes or related signaling pathways significantly contribute to cardiac dysfunction in this pathology. Thus, in the current review we aimed to discuss the expression and signaling pathways of β-AR in the diabetic heart. For this purpose, we have used ‘diabetes, heart, beta adrenoceptor’ and ‘diabetes, heart, beta adrenergic receptor’ keywords combination to search relevant articles in PubMed database. 

## 2. β-Adrenoceptors in the Diabetic Heart

Diabetes causes impaired cardiac function in which decreased β-AR mediated responsiveness has a major role [35]. This may at least partly be due to alterations in the expression of the β-AR subtypes and the signaling pathways they couple to. These effects of diabetes may differ between the β-AR subtypes. The contractile response mediated by β_1_- and β_2_-AR after stimulation with isoprenaline was reduced in streptozotocin (STZ)-induced diabetic rat papillary muscle [36,37,38,39]. On the other hand, β_3_-AR mediated relaxation was increased in the Langendorff perfused heart of STZ diabetic rats [40,41]. Similarly, diabetes has resulted in decreased expression of β_1_- and β_2_-AR whereas β_3_-AR have been found to be upregulated in this pathology [41,42,43]. 

Hereafter, we divide each of the three subsections mainly by type of diabetes and secondarily by species. As STZ injections are by far the most frequently applied model of T1DM, subsequent data on T1DM models always refer to that model unless explicitly noted otherwise.

### 2.1. mRNA and Protein Expression

Both mRNA and protein expression of β-AR subtypes in diabetes have been investigated, mostly in animals but to a more limited extent also in humans. Both T1DM and T2DM animal models have been used. Fewer experimental studies have been done in models of T2DM as compared to T1DM. This may have resulted from the fact that using T1DM models, particularly STZ injections is easier and requires less resources. Various species such as rodents, swine, dog, or hamster have been used. The duration of diabetes in such reports varied widely from 4 days to 13 months. 

The β-AR expression at the protein level has been studied mostly by using radioligand binding assays and immunoblot studies. β-AR mRNA expression has been determined by using PCR. In radioligand binding studies, [^3^H]-CGP 12,177, [^3^H]-dihydroalprenolol ([^3^H]-DHA), [^125^I]-iodocyanopindolol and [^125^I]-iodopindolol have been used as ligands for detection of β-AR. In most of these studies, except one [44], changes in β-AR protein density level have been given without any subtype distinction. However, in the concentration of the ligands used in these studies, they do not detect β_3_-AR [45]. Thus, radioligand binding studies effectively provide information on the regulation of β_1_- and β_2_-AR, but not of β_3_-AR. Unlike radioligand binding studies, immunoblotting is used to understand subtype selective regulation of β-AR. However, most of the commercial β-AR subtype antibodies are reported to have poor target selectivity [46]; accordingly reported findings may not be true reflections of the expression of β_3_-AR protein. These are important points that should be considered when interpreting the studies. In this section, firstly, changes in protein level (data obtained from radioligand binding and immunoblotting studies) and then changes in mRNA level are described separately in both T1DM and T2DM.

#### 2.1.1. mRNA and Protein Expression in Type 1 Diabetes Mellitus

The expression of β-AR in the diabetic rat heart has been first reported by Savarese and Berkowitz in 1979 [47]. They have demonstrated that the number of β-AR were 28% decreased in ventricular tissue from diabetic Sprague Dawley rats as assessed by [^3^H]-DHA binding 8 weeks after STZ injection. Such, downregulation of β-AR has been confirmed in many studies by using different methods at both protein and mRNA level (Table 1). As mentioned in the previous section, the decrease of the expression of β-AR subtypes has not been classified separately in most studies that performed radioligand binding assay to determine changes in receptor protein level [48,49,50,51,52,53,54,55,56,57,58,59,60,61,62,63,64,65,66,67,68,69,70,71]. However, unchanged β-AR density has been reported in female Wistar rats after 8-days of diabetes [72] and in 14-day diabetic male Wistar rats [73]. While it could be assumed that this result was related to very short duration of diabetes, similar findings have been reported after 6 weeks [69,74], 10 weeks [69], 12 weeks [75], 16 weeks [76], 90 and 200 days of diabetes in rats [77]. Similar β-AR density has also been reported in 16-week diabetic C57BL/6 mice compared to control group [78]. Moreover, one group even reported an increased expression in two consecutive studies in female rats, both 2 weeks [79] and 3 weeks after STZ injection [80].

Subtype specific alterations of β-AR have been also reported by using radioligand binding studies in combination with subtype-selective competitors. β_1_- and β_2_-AR density has been found to be downregulated in the AV node, whereas the expression of β_1_- and β_2_-AR were decreased and increased in interventricular septum, respectively in 3-week diabetic Wistar rats [44]. Apart from the studies on rodents, β-AR density in transmural left ventricle was not altered after 12 weeks of diabetes in male pigs [81] whereas the receptor number was found to be reduced in the right atrium of 11-week diabetic female swine [82]. β-AR density was decreased after 12 weeks of Alloxan-induced diabetes in New Zealand white rabbits [83] while no alteration was found after 10–13 months of diabetes in the same model [84]. β-AR density was reported to be increased after 3 and 14 weeks in Chinese spontaneously diabetic hamsters, but unchanged after 24 and 35 weeks [85].

Protein expression of β_1_-AR was found to be decreased in the diabetic rat heart by using immunoblots [37,38,39,41,42,43,86,87,88]. β_2_-AR protein expression level was shown to be downregulated in diabetic (Ins2+/− Akita) mice [89]. Protein expression of β_2_-AR was also reduced in 14-week STZ diabetic rats [42]. Similar findings have been presented by other study groups in the same model [41,43]. On the other hand, Sharma et al. have reported upregulation of β_2_-AR protein in 6-week STZ diabetic rat [87]. After the presence has been demonstrated in the cardiac tissue [12], the expressional status of β_3_-AR has been an issue of interest. Protein expression of β_3_-AR was almost doubled in 14-week STZ diabetic rat heart [42]. Of note, this was the first study to report the change of all subtypes in the diabetic rat heart. The upregulation of β_3_-AR in the diabetic rat heart has been confirmed in the studies in STZ-diabetic rat model [36,37,39,41,43,86,87,88,90].

Consistent with the reduced protein expression level, mRNA expression of β_1_-AR was decreased in the diabetic rat heart [36,41,42,91]. Despite downregulation of protein expression, β_2_-AR mRNA expression was found to be increased in 14-week STZ diabetic rats [42]. This finding has been supported by other investigators [41,43]. In line with the change in protein expression level, mRNA expression of β_3_-AR was shown to increased in 14-week STZ diabetic rat heart [42]. The upregulation of β_3_-AR mRNA level has also determined by others in STZ-diabetic rat model [40,41].

#### 2.1.2. mRNA and Protein Expression in Type 2 Diabetes Mellitus

β-AR protein level has been found to be unchanged after 4 and 12 weeks of diabetes in female db/db mice by using radioligand binding assay [92]. Similar β-AR protein level has been observed after 10–12 months of diabetes in a neonatal rat model of T2DM compared to control group [93]. Similarly, β-AR density was preserved in Zucker fa/fa obese rats at 6, 10, and 20 weeks of diabetes [69]. On the other hand, Dubois et al. showed that a reduced β-AR protein level in obese Zucker diabetic fatty (ZDF) rats [66].

β_1_-AR protein expression has found to be preserved in atrial appendages of diabetic patients [94]. However, high fat fed C57BL/6J mice did not present an alteration in the protein expression of β_1_-AR [95,96]. Protein expression of β_1_-AR was decreased in high fat fed-STZ injected rats while β_2_-AR density was not altered [97]. In another study, protein expression of both β_1_- and β_2_-AR were preserved whereas β_3_-AR were downregulated in 12-week diabetes induced by high fat/high sucrose diet and + %10 sucrose in drinking water [98]. However, Jiang et al. have demonstrated that β_1_- and β_2_-AR protein expression were reduced and β_3_-AR density was not altered ZDF rats compared to control [99]. Similar to these findings, β_1_- and β_2_-AR were shown to be downregulated whereas β_3_-AR density was preserved in high fat fed-STZ injected rats [100]. After this study, the same investigator group have determined the change of β-AR subtypes at 10 and 16 weeks of diabetes in ZDF rats. They have reported that protein expression of β_1_- AR was preserved at 10th week, however it was decreased at 16th week. On the other hand, β_2_- and β_3_-AR were downregulated and upregulated respectively at both time points [101]. In addition, expressional change of β-AR in left ventricle and right atrium was compared in 20-week old ZDF rats. The protein level of β_1_-AR and β_2_-AR was decreased and increased respectively in the left ventricle of diabetic animals whereas both β_1_- and β_2_-AR have been found to be upregulated in the right atrium [102].

The mRNA expression of β_1_- and β_2_-AR was decreased in the atrial appendage of diabetic patients [103]. Daniels et al. have reported that mRNA expression of β_1_-AR was not altered in both sexes in 10-week diabetic db/db mice [104].

### 2.2. β-AR Mediated Signaling Pathways

As explained in detail in Section 1, due to their G_s_- and/or G_i_-coupled structure, stimulation of the three subtypes of the β-AR in cardiac tissue leads to activation of different downstream pathways, resulting in contraction or relaxation of cardiomyocytes. In this section, we focus on the changes in these downstream pathways and in related molecules for G_s_- and G_i_-coupled β-AR mediated responses separately. 

#### 2.2.1. Changes in G_s_-Coupling β-Adrenoceptor Mediated Signaling Pathways in Diabetes

Stimulation of G_s_-coupled β_1_- and β_2_-AR results in activation of AC-cAMP-PKA pathway and later PKA-mediated phosphorylation of several key proteins which are responsible of contraction of cardiomyocytes. Similar [82] and decreased G_s_ expression level [81] in cardiac tissue has been shown in T1DM Yucatan micropigs and minipigs respectively. Uekita et al. reported that G_sα_ subunit expression did not differ T1DM hamsters compared to the control group [85]. Basal AC activity mostly remained unchanged in diabetic heart (Table 2), but increased basal AC activity has been shown in diabetes by some investigators [85,105]. Most studies have shown that basal cAMP level in cardiac tissue has not been changed in both T1DM [54,72,106,107,108,109,110,111] or T2DM [112] animals. Similarly, not cardiac but plasma cAMP level has not been found different in T1DM patients compared to healthy subjects [113,114]. Decreased cAMP levels were shown in long term T1DM [69] and T2DM animals [115]. In an insulin resistant rat model, basal cardiac cAMP level was found increased at 6 weeks, unchanged at 10 weeks and decreased at 20 weeks of diabetes [69]. Interestingly, Uekita et al. have found an increased basal cardiac cAMP level in hamsters [85]. While basal cAMP level remains mostly unchanged in diabetic heart, β-AR mediated cAMP accumulation has been found decreased in some [72,106,107,112,116] but not in all studies (Table 2).

Alteration in β-AR mediated AC activity in diabetes has been evaluated by using non-selective β-AR agonist isoprenaline. There are studies which show similar AC activity [50,54,74,81,86] or decreased AC activity [37,55,60,62,71,72,81,105,117,118,119] or increased AC activity [85] in response to β-AR agonist stimulation in T1DM animals. Schaffer et al. showed similar receptor-mediated AC activity in a non-insulin-dependent diabetes model [93]. Regardless of duration of diabetes, receptor-mediated AC activity has been found similar in both T1DM and T2DM diabetic rats compared to control group [69]. Bilginoglu et al. showed that β-AR mediated AC activity attenuated in male but remained unchanged in female T1DM rats [70]. Heterotrimeric G protein-dependent AC activity has been evaluated in studies of diabetes. Stimulation of heterotrimeric G protein has been shown to result in similar AC activity [50,62,117,118] or decreased AC activity [81,82,105,119] or increased AC activity [85] in diabetic animals compared to control group. Forskolin is a direct AC stimulator and often used to investigate intrinsic AC activity. Forskolin-induced AC activity was found similar in both T1DM [50,58,60,74,81,86,117] and T2DM animals [69] compared to the control group. On the other hand, attenuation of forskolin-induced AC activity in T1DM [37,82,105,120] and in T2DM [99] has been shown in other studies. Similarly, studies that found increased basal AC activity found increased AC activity in diabetes in response to forskolin [67,85]. Austin and Chess-Williams showed increased AC sensitivity to forskolin in T1DM [80]. 

Similar basal cardiac PKA activity has been shown in T1DM rats [87,110] and in T2DM mice [112]. Shao et al. also showed decreased basal PKA activity in T1DM rats [121]. Stimulation of PKA using by either 8-Bromo-cAMP or dibutyryl-cAMP mostly been shown to cause reduced activity in both T1DM and T2DM [37,99,110,120,122]. However, similar PKA activity was observed in diabetic animals in response to cAMP-derivative stimulation as in control animals [86,121]. In addition to this, isoprenaline stimulated PKA activity was decreased in T1DM rat [122] and T2DM mice [112]. While phosphorylation of PKA was not found different in 6-week T1DM rats [123], phospho-PKA/PKA ratio has been shown to increase when the duration of diabetes reached 12 week [75].

Changes in PKA-dependent phosphorylation of contractile proteins may also be responsible for diabetes related cardiac dysfunction. Alterations in PKA mediated phosphorylation of phospholamban (PLN), ryanodine receptor (RyR) and troponin I (TnI) proteins due to diabetes have been investigated in the studies which we used to generate this review. p-PLN expression was shown to decrease in T1DM rats [75]. Decreased p-PLN/PLN ratio was also found in T2DM animals [112,115]. p-PLN/PLN ratio has been shown to unchanged in T2DM mice, however isoprenaline stimulation caused a decrease in this ratio [96]. The p-RyR/RyR ratio was decreased in T1DM rats [75]. On the other hand, p-RyR expression at both Ser2814 and Ser2808 sites was found to increase in T1DM rats [121]. Whereas, decreased p-TnI/TnI ratio was shown in T2DM mice [112,115], unchanged p-TnI/TnI ratio was found by other research group [96] in the same diabetes model. While modulation of ion channel function, mostly various types of K^+^ and Ca^++^ channels, is an important effector pathway of cardiac β-AR, our search did not identify studies describing alterations of such coupling in diabetes; therefore, they are not discussed here. 

#### 2.2.2. Changes in G_i_-Coupling β-Adrenoceptor Mediated Signaling Pathways in Diabetes

Different from β_1_-AR, stimulation of β_2_-AR also activates the phosphatidylinositol 3-kinase (PI3K)/protein kinase B (AKT) pathway, which protects cardiac cells against apoptosis through its G_i_ coupling [124]. Unlike β_1_- and β_2_-AR, stimulation of β_3_-AR activates NOS-NO-cGMP-PKG pathway through its G_i_-coupling, resulting in relaxation of cardiomyocytes. It has been shown that the G_i_ expression level [82,96,115] and G_iα_ subunit expression level [85] do not change in diabetes. On the other hand, increased G_i_ expression level and G_i_/G_s_ ratio was found in T1DM Yucatan minipigs [81]. Kayki-Mutlu et al. have reported increased G_iα2_ subunit expression level in T1DM rats [40].

Coupling of cardiac β_2_-AR with the G_s_ and G_i_-protein did not differ between T1DM and control rats [87]. It has been shown that AKT phosphorylation [123], p-AKT protein expression [87] and p-AKT/AKT ratio [38] were reduced in T1DM animals. However, Wang et al. found an elevated p-AKT expression level in T2DM [115]. In T2DM, β_2_-AR activate cGMP-PKG signaling through a G_i_-coupled pathway, which has antihypertrophic effect on heart [112]. In this study, basal cardiac cGMP levels did not differ between diabetic and control animals, but an increased cGMP level has been observed in response to isoprenaline stimulation in diabetic animals [112]. 

There are few studies investigating changes in β_3_-AR mediated downstream molecules in diabetes. Basal cardiac cGMP levels were found to be unchanged in T1DM rats compared to their age-matched controls [106]. Increased NOS activity [37] and NOS1 protein expression were reported in T1DM rats [37,90]. Similarly, increased NO and cNOS expression level and iNOS induction were shown in T1DM rats [125]. However, eNOS expression did not change in T1DM rats after 8 weeks of diabetes [36,40]. Contrary to these findings, Kleindienst et. al. found increased eNOS and p-eNOS, both at Ser1177 and Thr495 site, expression in T2DM mice compared to the control group [98]. Moreover, it was found that nNOS expression decreased and iNOS expression increased in diabetic animals [98]. Consistent with this study, increased iNOS expression and activity were also found in genetically T2DM rats [126]. β_3_-AR mediated activation of Na^+^-K^+^ pump has beneficial effects on heart in pathological situation [127]. Unchanged Na^+^-K^+^ ATPase activity was found in T1DM hamsters [85]. However, Schaffer et al. showed that diabetes could cause reduction in Na^+^-K^+^ ATPase activity in long term [93].

### 2.3. The Inotropic and Chronotropic Response to β-AR Stimulation in the Diabetic Heart

The impact of diabetes on β-AR mediated cardiac function has been widely investigated as β-AR are the essential component of cardiac contraction [128]. The inotropic and chronotropic responses to β-AR stimulation in the diabetic heart has been determined both in vivo and in vitro. Cardiomyocytes, atrial or ventricular tissue or papillary muscle or whole heart preparations have been used in in vitro studied. The studies have mostly been conducted on rats but also been done on mice, swine, rabbit and humans both in T1DM and T2DM. 

In this section, β-AR functional changes in cardiac tissue is described, firstly, the contraction and/or relaxation responses obtained in in vitro studies and Ca^++^-mediated responses, if any, and then the responses obtained from in vivo studies in T1DM and T2DM separately.

#### 2.3.1. Type 1 Diabetes Mellitus

Ventricular myocytes have been isolated from Wistar rats and the isoprenaline-induced contractile response was not affected after 24 h of hyperglycemic exposure [129]. In line with this result, it has been shown that hyperglycemia has no detrimental effect on ventricular cardiomyocytes after isoprenaline stimulation [130]. The isoprenaline-mediated effect on peak shortening was reduced in ventricular myocytes isolated from STZ-diabetic mice [131]. Different from these studies, rate of contraction and relaxation time after isoprenaline treatment were augmented in cardiomyocytes of diabetic Ins2^+/−^Akita mice [89].

The inotropic effect of isoprenaline was attenuated in chronic diabetic rat atria [132,133,134]. Reduced inotropic effect in right atria has been also confirmed despite absence of a relevant change in the chronotropic effect [135]. Decreased inotropic effect to β-AR stimulation has also been observed in the left atria [136]. Reduced chronotropic effect of β-AR agonists has been reported by several investigators [132,133,137,138]. While isoprenaline stimulated chronotropic response has been found to be impaired in right atria of acute and chronic diabetes [51], isoprenaline induced inotropy has mostly been found unaltered in the studies by same study group [56,57,58,139] except for the one study with enhanced inotropic response [51]. The idea of increased inotropic effect of isoprenaline has been supported in the left atria of 2-week diabetic rats which was reversed after 12 weeks of the pathology [140]. Preserved inotropic effect of β-AR stimulation has been also shown in the right and left atria of spontaneously diabetic Bio-breeding (BB) rats [61]. Isoprenaline induced chronotropic effect of the right atrium was not different in diabetic Yucatan minipigs [82], whereas it was found to be depressed in diabetic rats [141].

The discrepancy between the functional studies are also present in the isolated ventricular tissue. A reduced maximum response to isoprenaline in right ventricular strips [142] and decreased contractile response to isoprenaline in ventricular tissue [143] have been observed in diabetic rats. Isoprenaline-induced developed tension of right ventricular strip was numerically decreased in alloxan diabetic rabbit, but this effect has not been found to be statistically significant [84]. Gunasekaran et al. have demonstrated that isoprenaline induced maximum response was not altered in right ventricular strip of 4-week diabetic rats [64]. On the other hand, Wald et al. have demonstrated that force of contraction was increased in the ventricle of STZ diabetic rats [144]. β-AR responsiveness has been also investigated by using papillary muscles. Contractile force generation by isoprenaline stimulation was reduced in the papillary muscle in 8-week diabetic rats [48]. This finding has been confirmed by other groups [37,38,39,145,146,147,148,149,150] including our group [36]. On the other hand, preserved contractile response to isoprenaline in the papillary tissue has also been reported in long term diabetes [79]. Preserved positive lusitropic effect in response to isoprenaline administration, in the presence of β_3_-AR antagonist and NOS inhibitor, in papillary muscle has also been reported in both acute and chronic diabetes [86]. We showed blunted relaxation response to BRL 37344, a mixed β_2_/β_3_-AR agonist, in the papillary muscle [36]. The sensitivity to noradrenaline in right atria or ventricular papillary muscle was not changed in diabetic rabbits [151]. However, Austin et al. showed increased sensitivity to isoprenaline of both left atria and papillary muscle without any significant alteration in the maximum response to isoprenaline in 14-day diabetic female rat [152]. 

Whole heart preparation has been used in some of the studies. The peak response to isoprenaline stimulation was not changed in Langendorff perfused hearts in 9-week diabetic rats [88]. Maximum inotropic response to isoprenaline was found to be increased in female rat heart while it was not changed in male rats [70]. Preserved responses to β-AR agonists were also found by others [153,154]. Comparable results in maximum increase in developed tension, heart rate, +dp/dt and -dp/dt after isoprenaline stimulation were observed among the groups although the increase at submaximal doses of the drug was greater in the diabetic group [73]. On the other hand, using Langendorff or working heart preparation, an impaired response to β-AR stimulation was also shown [71,106,125,155,156,157]. Unchanged +dp/dt in response to isoprenaline stimulation was shown by using working heart preparation in both alloxan and STZ diabetic female rats. However, -dp/dt was depressed at both acute and chronic phase of diabetes in the same study [158]. In some of the studies, the functional response to β-AR stimulation has been subtype specifically investigated. Dobutamine induced β_1_-AR mediated inotropic effect was enhanced in the diabetic heart whereas salbutamol-induced β_2_-AR mediated contractility was comparable in STZ diabetic spontaneously hypertensive (SHR) rats compared to control group [67]. β_1_- and β_2_-AR mediated inotropic effect was preserved and reduced in STZ diabetic rat heart, respectively. On the other hand, BRL 37344 stimulated β-AR mediated relaxation was increased in Langendorff perfused hearts [41]. Augmented relaxation response to BRL 37344 in the same diabetic model has been also shown by our group [40].

β-AR mediated responses have been also evaluated by the change in Ca^++^ transients. Increase in Ca^++^ transient amplitude as a response to orciprenaline was lower in cardiomyocytes isolated from 1- and 6-week diabetic rats [159]. Isoprenaline induced effect in Ca^++^ transient and cell shortening was impaired in ventricular myocytes isolated from 4–6 week diabetic rats [120]. Parallel to these findings, blunted [Ca^++^]_i_ change in response to isoprenaline was reported in diabetic cells [122].

There are relatively fewer studies which have investigated β-AR mediated in vivo in T1DM. In vivo cardiac parameters such as rate of contraction/relaxation and heart rate were attenuated after isoprenaline treatment in STZ diabetic mice [110]. Similarly, isoprenaline induced in vivo inotropic response was decreased in 16-week diabetic mice [78]. +dp/dt was depressed after isoprenaline stimulation in 4-week diabetic rats whereas it was preserved in 2-week diabetic group [55]. Both +dp/dt and -dp/dt and amplitude of response were reduced after in vivo isoprenaline administration in 7-week diabetic rats [43]. Decreased response to in vivo β-AR stimulation in the diabetic rat heart has been shown also by others [37,39,75,91,160,161,162,163,164,165,166]. Depressed isoprenaline stimulated response has also been shown in alloxan diabetic rabbits [83]. However, unaltered or enhanced inotropic/lusitropic response to in vivo β-AR stimulation have been also reported. Amour et al. have demonstrated that dobutamine induced positive lusitropic effect was not changed after 4 or 12 weeks of diabetes despite diastolic dysfunction [86]. Similarly, in vivo β-AR induced effect was well preserved in female diabetic rats [167]. On the other hand, dobutamine induced contractile function was found to be increased in diabetic rats [168]. However, the effect of dobutamine on in vivo cardiac parameters were comparable between control and STZ-diabetic Yucatan minipigs [169]. There are also few studies which have determined the change in β-AR mediated effect in the human heart. β-AR sensitivity was increased in insulin dependent diabetic patients [170]. On the other hand, it has been found that sensitivity to isoprenaline stimulation was reduced in insulin dependent diabetic patients with hypoglycemic unawareness [114,171,172]. However, epinephrine induced increase in heart rate was demonstrated to be greater in diabetic patients with autonomic neuropathy [113]. On the other hand, the effect of in vivo noradrenaline administration on cardiac parameters was comparable in control and diabetic patients [173]. 

#### 2.3.2. Type 2 Diabetes Mellitus

As mentioned in the previous sections, T2DM animal models have received attention because of their greater translational value. Parallel to the studies in T1DM models, mostly rats, but also mice have been used in studies on T2DM. 

Isoprenaline-stimulated sarcomere shortening in cardiomyocytes isolated from high fat fed mice was not found different compared to control cardiomyocytes [174]. Isoprenaline induced chronotropic response was significantly increased in atria from fructose and sucrose fed rats which presents an insulin resistance model [175]. β-AR mediated cardiac function in right atria from was not found different among diabetic and healthy subjects [138]. 

The inotropic effect of isoprenaline was not altered in left ventricular trabecular muscle in neonatal non-insulin-dependent diabetes model [143]. Similarly, trabeculae tissue was found to be unresponsive to dobutamine stimulation in T2DM individuals [94]. Positive inotropic effect of isoprenaline was slightly and markedly decreased in papillary muscle from Zucker obese and ZDF rats, respectively [99].

β-AR mediated response on contraction and relaxation was impaired in ZDF rats by using the Langendorff heart preparation. It has been suggested that β_1_-AR is the main subtype which regulates heart function in both healthy and diabetic rats whereas β_2_-AR has an indirect influence on β-AR mediated responses in the diabetic heart [176]. Dobutamine induced chronotropic effect was reduced in unpaced hearts despite of preserved inotropic effect in whole heart in ZDF rats. Of note, contractility in diabetic groups was decreased when the heart rate is set to 300 bpm with pacing [102]. Impaired contraction and relaxation response to isoprenaline stimulation has been also shown in isolated perfused heart in neonatal noninsulin dependent diabetes model [93]. Left ventricular developed pressure after BRL 37344 stimulation was determined to be significantly increased by using Langendorff heart preparation in high fat fed mice [98].

The in vivo effect of isoprenaline was attenuated in high fat fed rats [96,112,115]. The in vivo inotropic response to dobutamine stimulation was reduced in ZDF rats [102]. Similar findings were reported by Song et al. in the same model [126]. In line with these findings, in vivo inotropic and lusitropic effect of dobutamine was found to be reduced in db/db mice [104]. However, Takada et al. have demonstrated that positive inotropic response to in vivo β-AR stimulation was similar between control and OLETF diabetic rats [177]. This finding was confirmed in diabetic individuals. Both chronotropic and inotropic response to β-AR stimulation was found to be unaffected in this study [178]. Chronotropic response to dobutamine stimulation was decreased in conscious ZDF rats [179]. Chronotropic effect of isoprenaline was augmented in ZDF rats and it has been implicated that β_1_-AR is the main subtype to modulate chronotropic effect in control and diabetic animals [180]. Furthermore, heart rate was not altered in db/db diabetic mice after a 2-h isoprenaline administration [111].

## 3. Discussion

Cardiovascular complications are the major risk factor for mortality in individuals with diabetes [181]. It is important to understand underlying mechanisms that contributes to diabetes-induced cardiac dysfunction to allow researchers and physicians to prevent or treat it. Sympathetic system overactivity is well-known characteristic of diabetes. Thus, it seems logical that sympathetic overdrive leads changes in β-AR mediated responses such as cardiac contraction and relaxation. This is an issue of interest which has been investigated by several study groups for many years. However, there are many inconsistent findings in the literature which makes it difficult to properly interpret the existing data. Moreover, the literature is dominated by studies from animal models, which may or may not be representative for the human situation. The conflicting results from the animal studies make it even harder to extrapolate to humans. Therefore, more human studies are urgently needed. Two additional under-investigated areas emerged from our search: There is little data on alterations of β-AR coupling to ion channels in the diabetic heart and on differential regulation within the heart, for instance ventricular tissues vs. conduction system.

Most investigators have reported a decreased expression of β-AR by using radioligand binding studies in rodent models of T1DM, but a smaller number of studies did not confirm such changes, and an increased expression has been reported in two studies from one group of investigators [79,80]. The balance of these findings indicates that expression is decreased, primarily due to a reduction in β_1_- and β_2_-AR and less if any in β_3_-AR as explained in Section 2.1. Similarly to radioligand binding studies, there are different findings about changes in subtype specific changes in β-AR by using immunoblotting by different study groups. However, changes in same subtype could differ regarding the different cardiac tissue that is used [102] or duration of diabetes [101] in the same study. There are two studies which have reported both protein and mRNA level of β-AR in diabetic heart [41,42]. In both studies, the mRNA expression of β_1_-AR and β_3_-AR was found to be decreased and increased respectively, consistent with the protein level. However, it was found that the β_2_-AR mRNA level was increased despite the reduced β_2_-AR protein level, and this has been referred to the sensitization of the receptor [42]. While not confirmed in T2DM models, an increased expression of β_3_-AR is one of the most consistent findings in the STZ model of T1DM. Similarly, upregulation of β_3_-ARs have been reported in the heart failure. In both human [29] and canine cardiomyocytes [182], the expression of β_3_-ARs was found to be increased. On the other hand, these studies have indicated conflicting results about the β_3_-AR mediated relaxation in the failed heart. The response to β_3_-AR stimulation was blunted in the human heart [29] while it was increased in the canine heart [182]. It has been suggested that this discrepancy may have been caused by several factors such as interspecies differences, the severity of the disease, tissue type or different measurement technique of the contractility [182]. β_3_-ARs have been also related to a protective role in cardiac ischemia and reperfusion injury. The beneficial effects due to β_3_-AR stimulation have been demonstrated in this pathology [183,184]. Of note, the favorable effect mediated by β_3_-AR stimulation in ischemia and reperfusion injury was seen only in control rats not in high fat high sucrose (HFS) fed mice [98]. This result has been linked to reduced expression of β_3_-ARs in HFS mice. These findings imply that the expressional or functional changes of β_3_-ARs may contribute to the cardiac pathologies. 

Cardiac β-AR post-receptor signaling pathway regulation may contribute to changes in β-AR responsiveness in diabetes. Supporting that, reduced β-AR responsiveness has been reported despite unchanged β-AR protein and/or mRNA level in diabetic state in some studies [96,115]. However, there are few studies which have linked the changes in β-AR subtype in molecular level and related downstream molecules and receptor mediated functional response [36,40,96,115]. Some studies have investigated the changes in the protein expression of β-AR and concomitant signaling pathway molecules, with [60,82] or without [54,81] functional response studies. And some of them have evaluated changes in signaling pathway molecules and further contraction and/or relaxation responses [110,111]. Studies investigating the cardiac β-AR signaling pathway have also revealed quite different results (Table 2). Conflicted findings have been reported on the same downstream molecule in the studies which have been done by same study group with the very same experimental approach despite similar findings on β-AR protein expression changes [37,86]. 

Same as the molecular changes in β-AR, in vivo and in vitro studies have yielded different results. Increased and decreased response to the same agonist, in the whole heart [40] and in the papillary muscle [36], respectively, was also shown by our study group. It has been suggested in the studies that discrepancies in findings may have resulted from experimental approach such as duration of the diabetes or the cardiac section that was used in the study. However, to interpret existing data by attributing to single or several variables cannot go beyond an assumption, considering the existence of such inconsistent and different findings.

In conclusion, since the late 1970s, diabetes related changes in cardiac β-AR have attracted attention of several researchers but their findings are controversial. At the molecular level, antibody-based approaches are troubled by low target selectivity and in general studies have a very poor reproducibility rate. The studies primarily have been conducted on rodent cardiac tissues and few have been done in other species including humans. In preclinical studies, mostly male animals have been preferred, there are relatively fewer studies which have been done in female animals. Even though 90% of diabetic patients have T2DM [185], more studies have been done on T1DM animal models. Apart from inconsistent findings, these are other obstacles to interpret and generalize findings to the clinical stage. Thus, it is unlikely to make a general interpretation and extrapolation regarding to the studies which have been done so far. We think that investigators who do or will work on cardiac β-AR in diabetes should be aware of the inconsistency in the literature. More studies are needed to come through an overall conclusion about the role of β-AR in diabetic cardiac dysfunction.

## Figures and Tables

**Table 1 cells-09-02548-t001:** Cardiac β-Adrenoceptors (β-AR) protein and mRNA levels in type 1 (T1DM) and type 2 diabetes (T2DM).

Reference	β-AR Protein (Binding)	β-AR Protein (Western Blot)	β-AR mRNA	Species	Sex	Diabetes Model	Duration of Diabetes
Amour et al., 2007	n/a	β_1_-AR ↓ β_3_-AR ↑	n/a	Wistar rat	Male	STZ induced T1DM	4-week
Amour et al., 2008	n/a	β_1_-AR ↓ β_3_-AR ↑	n/a	Wistar rat	Male	STZ induced T1DM	4- and 12-week
Aragno et al., 2012	n/a	β_1_-AR ↓	n/a	Wistar rat	Male	STZ induced T1DM	6-week
Arioglu-Inan et al., 2013	n/a	β_3_-AR ↑	β_1_-AR ↓	SD rat	Male	STZ induced T1DM	8-week
Atkins et al., 1985	β-AR ↓	n/a	n/a	SD rat	Male	STZ induced T1DM	2- and 4-week
Austin and Chess-Williams, 1991	β-AR ↑	n/a	n/a	Wistar rat	Female	STZ induced T1DM	3-week
Austin and Chess-Williams, 1992	β-AR ↑	n/a	n/a	Wistar rat	Female	STZ induced T1DM	2-week
Beenen et al., 1997	β-AR ↓	n/a	n/a	SHR rat, WKY rat	Male	STZ induced T1DM	8-week
Bidasee et al., 2008	n/a	β_1_-AR ↓ β_2_-AR ↓ β_3_-AR ↑	n/a	SD rat	Male	STZ induced T1DM	7-week
Bilginoglu et al., 2007	β-AR binding site ↓	n/a	n/a	Wistar rat	Male Female	STZ induced T1DM	5-week
Bilginoglu et al., 2009	β-AR ↓	n/a	n/a	Wistar rat	Male	STZ induced T1DM	5-week
Bitar et al., 1987	β-AR ↓	n/a	n/a	SD rat	Male	STZ induced T1DM	2-month
Carillion et al., 2017	n/a	β_1_-AR ↓ β_3_-AR ↑	n/a	Wistar rat	Male	STZ induced T1DM	8-week
Cros et al., 1986	β-AR n.c.	n/a	n/a	Rat	n/a	STZ induced T1DM	4-month
Dincer et al., 2001	n/a	β_1_-AR ↓ β_2_-AR ↓ β_3_-AR ↑	β_1_-AR ↓ β_2_-AR ↑ β_3_-AR ↑	Wistar rat	Male	STZ induced T1DM	14-week
Dubois et al., 1996	β-AR ↓	n/a	n/a	SHR rat, WKY rat	Male	STZ induced T1DM	8-week
Durante et al., 1989	β-AR ↓	n/a	n/a	Spontaneously diabetic Bio-Breeding (BB) rats	n/a	Genetic T1DM	10-week
Eckel et al., 1991	β-AR ↓	n/a	n/a	Wistar rat	Male	STZ induced T1DM	3-week
Gotzsche, 1983	β-AR n.c.	n/a	n/a	Wistar rat	Female	STZ induced T1DM	8-day
Gunasekaran et al., 1993	β-AR ↓	n/a	n/a	SD rat	Male	STZ induced T1DM	4-week
Heyliger et al., 1982	β-AR ↓	n/a	n/a	SD rat	Male	STZ induced T1DM	8-week
Huisamen et al., 2001	β-AR n.c. (6 and 10-week) β-AR ↓ (20-week)	n/a	n/a	Wistar rat	n/a	STZ induced T1DM	6, 10- and 20-week
Ingebretsen et al., 1983	β-AR ↓	n/a	n/a	Albino SD rat	Male	Alloxan induced T1DM	5-day
Kayki-Mutlu et al., 2014	n/a	n/a	β_3_-AR ↑	SD rat	Male	STZ induced T1DM	8-week
Lahaye Sle et al., 2010	n/a	β_1_-AR ↓ β_2_-AR n.c. β_3_-AR ↑	n/a	Wistar rat	Male	STZ induced T1DM	9-week
Latifpour and McNeill, 1984	β-AR ↓	n/a	n/a	Rat	n/a	STZ induced T1DM	6-month
Le Douairon Lahaye et al., 2011	n/a	β_3_-AR ↑	n/a	Wistar rat	Male	STZ induced T1DM	9-week
Lee et al., 2004	β-AR ↓	n/a	n/a	New Zealand white rabbit	Male	Alloxan induced T1DM	12-week
Matsuda et al., 1999	β-AR ↓	n/a	β_1_-AR ↓	Wistar rat	Male	STZ induced T1DM	6-week
Mishra et al., 2010	n/a	β_2_-AR ↓	n/a	(Ins2+/− Akita) mice	Male	Genetic T1DM	12-week
Monnerat-Cahli et al., 2014	n/a	n/a	β_1_-AR ↓	Wistar rat	Male	STZ induced T1DM	8-week
Mooradian et al., 1988	β-AR n.c.	n/a	n/a	CDF (F-344) rat	Male	STZ induced T1DM	6-week
Myers et al., 2016	β-AR n.c.	n/a	n/a	C57BL/6 mice	Male/Female	STZ induced T1DM	16-week
Nishio et al., 1988	β-AR ↓	n/a	n/a	SD rat	Male	STZ induced T1DM	1-, 3- and 10-week
Okatan et al., 2015	n/a	β_1_-AR ↓ β_2_-AR ↓ β_3_-AR ↑	β_1_-AR ↓ β_2_-AR ↑ β_3_-AR ↑	Wistar rat	Male	STZ induced T1DM	n/a
Plourde et al., 1991	β-AR ↓	n/a	n/a	Wistar rat	Male	STZ induced T1DM	10-day + 10-week
Ramanadham et al., 1983	β-AR ↓	n/a	n/a	SD rat	Male	STZ induced T1DM	4-week
Ramanadham and Tenner, 1983	β-AR ↓	n/a	n/a	SD rat	Male	STZ induced T1DM	4-week
Ramanadham and Tenner, 1986	β-AR ↓	n/a	n/a	SD rat	Male	STZ induced T1DM	1-,3- and 6-month
Ramanadham and Tenner, 1987	β-AR ↓	n/a	n/a	SD rat	Male	STZ induced T1DM	4-week
Ramanadham et al., 1987	β-AR ↓	n/a	n/a	SD rat	Male	STZ induced T1DM	4-week
Roth et al., 1995	β-AR n.c.	n/a	n/a	Yucatan minipig	Male	STZ induced T1DM	12-week
Savarese and Berkowitz, 1979	β-AR ↓	n/a	n/a	SD rat	Male	STZ induced T1DM	8-week
Saito et al., 1991	β-AR ↓ (AV node) β-AR ↓ (IVS) β_1_-AR ↓ (AV node) β_1_-AR ↓ (IVS) β_2_-AR ↓ (AV node) β_2_-AR ↑ (IVS) β_1_/β_2_-AR (%) ↓ (AV node) β_1_/β_2_-AR ↓ (%) (IVS)	n/a	n/a	Wistar rat	Male	STZ induced T1DM	3-week
Sellers and Chess-Williams, 2001	β-AR n.c.	n/a	n/a	Wistar rat	Male	STZ induced T1DM	14-day
Sharma et al., 2008	n/a	β_1_-AR ↓ β_2_-AR ↑ β_3_-AR ↑	n/a	Wistar rat	Male	STZ induced T1DM	6-week
Sun et al., 2016	n/a	β_1_-AR n.c.	n/a	SD rat	Male	STZ induced T1DM	8-week
Sundaresan et al., 1984	β-AR ↓	n/a	n/a	SD rat	Male	STZ induced T1DM	8-week
Stanley et al., 2001	β-AR ↓	n/a	n/a	Yucatan micropig	Female	STZ induced T1DM	11-week
Sylvestre-Gervais et al., 1984	β-AR ↓	n/a	n/a	Wistar rat	Male	STZ induced T1DM	10-week
Takeda et al., 1996	β-AR n.c. (15, 18, 21, 24-day) β-AR ↓ (27-day)	n/a	n/a	SD rat	Male	STZ induced T1DM	15, 18, 21, 24 and 27-day
Tuncay et al., 2013	β-AR n.c.	n/a	n/a	Wistar rat	Male	STZ induced T1DM	12-week
Uekita et al., 1997	β-AR ↑ (3- and 14-week) β-AR n.c. (24- and 35-week)	n/a	n/a	CHAD hamsters	Male/Female	Genetic T1DM	3-, 14-, 24- and 35-week
Williams et al., 1983	β-AR ↓	n/a	n/a	Wistar rat	Male	STZ induced T1DM	8-week
Zola et al., 1988	β-AR n.c.	n/a	n/a	New Zealand white rabbit	Male	Alloxan induced T1DM	10–13 months
Daniels et al., 2010	n/a	na	β_1_-AR n.c.	db/db mice	Male/Female	Genetic T2DM	10-week
Dincer et al., 2003	n/a	n/a	β_1_-AR ↓ β_2_-AR ↓	Human	Male/Female	T2DM	<5-year
Dubois et al., 1996	β-AR ↓	n/a	n/a	Zucker obese rat	n/a	Insulin resistant diabetes	20-week
Fu et al., 2017	n/a	β_1_-AR n.c β_2_-AR n.c.	n/a	C57BL/6 mice	Male	HFD induced T2DM	8-week
Garris, 1990	β-AR n.c.	n/a	n/a	db/db mice	Female	Genetic T2DM	4- and 12-week
Haley et al., 2015	β-AR ↓	β_1_-AR ↓ β_2_-AR ↓ β_3_-AR n.c.	n/a	SD rat	Male	HDF/low dose STZ induced T2DM	8-week
Haley et al., 2015	β-AR n.c. (10-week) β-AR ↓ (16-week)	β_1_-AR n.c. (10-week) β_1_-AR ↓ (16-week) β_2_-AR ↓ (10-/16-week) β_3_-AR ↑ (10-/16-week)	n/a	ZDF rat	Male	Genetic T2DM	10- and 16-week
Huisamen et al., 2001	β-AR n.c.	n/a	n/a	Zucker obese rat	n/a	Insulin resistant diabetes model	6-, 10- and 20-week
Jiang et al., 2015	n/a	β_1_-AR ↓ β_2_-AR ↓ β_3_-AR n.c.	n/a	Zucker obese diabetic rat	Male	Genetic T2DM	15-week
Kleindienst et al., 2016	n/a	β_1_-AR n.c. β_2_-AR n.c. β_3_-AR ↓	n/a	C57BL/6 mice	Male	HF/HS diet induced T2DM	12-week
Lamberts et al., 2014	n/a	β_1_-AR n.c.	n/a	Human	Male/Female	T2DM	<1-year
Schaffer et al., 1991	β-AR n.c.	n/a	n/a	Neonatal Wistar rat	Male	Non insulin dependent diabetes	10- and 12-month
Thackeray et al., 2011	β-AR n.c (2-week) β-AR ↓ (8-week)	β_1_-AR ↓ (8-week) β_2_-AR n.c. (8-week)	n/a	SD rat	Male	HDF/low dose STZ induced T2DM	2- and 8-week
Thaung et al., 2015	n/a	β_1_-AR ↓ (LV) β_1_-AR ↑ (RA) β_2_-AR ↑ (LV and RA)	n/a	ZDF rat	Male	Genetic T2DM	20-week
Wang et al., 2017	n/a	β_1_-AR n.c.	n/a	C57BL/6J mice	Male	HFD induced T2DM	6-month

AV: atrioventricular, HF: high fat, HFD: high fat diet, HS: high sucrose, IVS: interventricular septum, LV: left ventricle, RA: right atrium, SD: Sprague Dawley, SHR: spontaneously hypertensive, STZ: streptozotocin, T1DM: type 1 diabetes mellitus, T2DM: type 2 diabetes mellitus, WKY: Wistar Kyoto, ZDF: Zucker diabetic fatty. n.c.: no change, n/a: no data available, ↓: decreased, ↑: increased.

**Table 2 cells-09-02548-t002:** The change in cardiac β-AR signaling pathway in T1DM and T2DM.

Reference	Downstream Molecule	Change	Species	Sex	Diabetes Model	Duration of Diabetes
Amour et al., 2007	G protein catalytic subunit dependent AC activity Receptor mediated AC activity Stimulated PKA activity NOS activity NOS1 expression	↓ ↓ ↓ ↑ ↑	Wistar rat	Male	STZ induced T1DM	4-week
Amour et al., 2008	G protein catalytic subunit dependent AC activity Receptor mediated AC activity Stimulated PKA activity	n.c. n.c. n.c.	Wistar rat	Male	STZ induced T1DM	4-week
Aragno et al., 2012	p-AKT/AKT	↓	Wistar rat	Male	STZ induced T1DM	6-week
Arioglu-Inan et al., 2013	eNOS expression	n.c.	SD rat	Male	STZ induced T1DM	8-week
Austin and Chess-Williams, 1991	G protein catalytic subunit dependent AC sensitivity	↑	Wistar rat	Female	STZ induced T1DM	3-week
Atkins et al., 1985	Receptor mediated AC activity	↓	SD rat	Male	STZ induced T1DM	4-week
Beenen et al., 1997	G protein catalytic subunit dependent AC activity	↑ (SHR diabetic rat)	SHR ratWKY rat	Male	STZ induced T1DM	8-week
Bilginoglu et al., 2007	Receptor mediated AC activity Receptor mediated AC activity	↓ (male) n.c. (female)	Wistar rat	Male/Female	STZ induced T1DM	5-week
Bilginoglu et al., 2009	Receptor mediated AC activity	↓	Wistar rat	Male	STZ induced T1DM	5-week
Bockus and Humphries, 2015	Basal cAMP level Basal PKA activity Stimulated PKA activity	n.c. n.c. ↓	C57BL/6J mice	Male	STZ induced T1DM	4-month
Das, 1973	Basal cAMP level Basal AC activity	n.c. n.c.	SD rat	Male	STZ induced T1DM	7-day
El-Hage et al., 1985	Basal cAMP level Receptor mediated cAMP level	n.c. n.c.	CDI miceC57BL/Ksjj mice	Male Male	Alloxan induced T1DM Genetic diabetes	10-day 10-day
Gotzsche, 1983	Basal cAMP levels Receptor mediated cAMP levels Receptor mediated AC activity	n.c. ↓ ↓	Wistar rat	Female	STZ induced T1DM	8-day
Huisamen et al., 2001	Basal AC activity G protein catalytic subunit dependent AC activity Receptor mediated AC activity Basal AC activity G protein catalytic subunit dependent AC activity Receptor mediated AC activity Basal AC activity G protein catalytic subunit dependent AC activity Receptor mediated AC activity Basal cAMP level Receptor mediated cAMP level Basal cAMP level Receptor mediated cAMP level Basal cAMP level Receptor mediated cAMP level	n.c. (6-week) n.c. (6-week) n.c. (6-week) n.c. (10-week) n.c. (10-week) n.c. (10-week) n.c. (20-week) n.c. (20-week) n.c. (20-week) n.c. (6-week) n.c. (6-week) n.c. (10-week) n.c. (10-week) ↓ (20-week) n.c. (20-week)	Wistar rat	n/a	STZ induced T1DM	6-, 10- and 20-week
Ingebretsen Jr et al., 1981	Basal cAMP level Basal cGMP level Receptor mediated cAMP level	n.c. n.c. ↓	Albino SD rat	Male	Alloxan induced T1DM	n/a
Ingebretsen et al., 1983	Basal AC activity G protein catalytic subunit dependent AC activity G protein dependent AC activity Receptor mediated AC activity	n.c. n.c. n.c. n.c.	SD rat	Male	Alloxan induced T1DM	5-day
Kayki-Mutlu et al., 2014	Giα2 expression eNOS expression	↑ n.c.	SD rat	Male	STZ induced T1DM	8-week
Le Douairon Lahaye et al., 2011	NOS1 expression	↑	Wistar rat	Male	STZ induced T1DM	9-week
Menahan et al., 1977	Basal AC activity G protein dependent AC activity Receptor mediated AC activity	n.c. ↓ ↓	Rat	n/a	Alloxan induced T1DM	13–14 days
Michel et al., 1985	Basal AC activity G protein dependent AC activity Receptor mediated AC activity	n.c. n.c. ↓	Wistar rat	Male	STZ induced T1DM	4-month
Miller Jr, 1984	Basal cAMP level Receptor mediated cAMP level	n.c. ↓	SD rat	Male	Alloxan induced T1DM	3–7 days
Mooradian et al., 1988	G protein catalytic subunit dependent AC activity Receptor mediated AC activity	n.c. n.c.	CDF (F-344) rat	Male	STZ induced T1DM	6-week
Nishio et al., 1988	Basal AC activity G protein catalytic subunit dependent AC activity Receptor mediated AC activity	n.c. n.c. ↓	SD rat	Male	STZ induced T1DM	10-week
Plourde et al., 1991	Basal AC activity G protein dependent AC activity Receptor mediated AC activity	n.c. n.c. ↓	Wistar rat	Male	STZ induced T1DM	10-day + 10-week
Ramanadham and Tenner, 1987	G protein catalytic subunit dependent AC activity	n.c.	SD rat	Male	STZ induced T1DM	4-week
Roth et al., 1995	Basal AC activity G protein catalytic subunit dependent AC activity G protein dependent AC activity Receptor mediated AC activity Gs expression Gi expression Gi/Gs	n.c. n.c. ↓ ↓ ↓ ↑ ↑	Yucatan minipig	Male	STZ induced T1DM	12-week
Sharma et al., 2008	Basal PKA activity p-AKT expression β2-Gs couplingβ2-Gi coupling	n.c. ↓ n.c. n.c.	Wistar rat	Male	STZ induced T1DM	6-week
Sharma et al., 2011	PKA phosphorylation AKT phosphorylation	n.c. ↓	Wistar rat	Male	STZ induced T1DM	6-week
Shao et al., 2009	Basal PKA activity Stimulated PKA activity p-RyR2 (Ser^2814^) expression p-RyR2 (Ser^2808^) expression	↓ n.c. ↑ ↑	SD rat	Male	STZ induced T1DM	7-week
Smith et al., 1984	Basal AC activity G protein catalytic subunit dependent AC activity G protein dependent AC activity Receptor mediated AC activity	n.c. n.c. n.c. ↓	SD rat	Male	STZ induced T1DM	8–9 weeks
Smith et al., 1997	NO expression cNOS expression iNOS induction	↑ ↑ ↑	SD rat	Male	STZ induced T1DM	8-week
Srivastava and Anand-Srivastava, 1985	Basal AC activity G protein catalytic subunit dependent AC activity G protein dependent AC activity Receptor mediated AC activity	↑ ↓ ↓ ↓	SD rat	Male	STZ induced T1DM	5-day
Stanley et al., 2001	Basal AC activity G protein catalytic subunit dependent AC activity G protein dependent AC activity Receptor mediated AC activity Gs expression Gi expression	n.c. ↓ ↓ n.c. n.c. n.c.	Yucatan micropig	Female	STZ induced T1DM	11-week
Sundaresan et al., 1984	Basal cAMP levels Receptor mediated cAMP levels Receptor mediated AC activity	n.c. n.c. n.c.	SD rat	Male	STZ induced T1DM	2-month
Tamada et al., 1998	G protein catalytic subunit dependent AC activity Stimulated PKA activity	↓ ↓	Wistar rat	Male	STZ induced T1DM	4–6 weeks
Trovik et al., 1994	Plasma cAMP level	n.c.	Human	Male/Female	Insulin dependent T1DM	n/a
Tuncay et al., 2013	p-PKA/PKA p-PLN expression p-RyR/RyR	↑ ↓ ↑	Wistar rat	Male	STZ induced T1DM	12-week
Uekita et al., 1997	Basal AC activity G protein catalytic subunit dependent AC activity G protein dependent AC activity Receptor mediated AC activity Na+-K+ ATPase activity Basal cAMP levelGiα expression Gsα expression	↑ (14-week) ↑ (14-/24-week) ↑ (14-/24-week) ↑ (14-/24-week) n.c.↑ (14-week) n.c. n.c.	CHAD hamsters	Male/Female	Genetic T1DM	3-, 14-, 24- and 35-week
Vadlamudi and McNeill, 1983	Basal cAMP level Receptor mediated cAMP level Basal cAMP level Receptor mediated cAMP level	n.c. (3-day) n.c. (3-day) n.c. (100–120 days) n.c. (100–120 days)	Wistar rat	Female	STZ and/or alloxan induced T1DM	3-day, 100–120 days
Wichelhaus et al., 1994	Receptor mediated cAMP level	↓	Wistar rat	Male	STZ induced T1DM	4–5 weeks
Yu et al., 1994	Stimulated PKA activity Receptor mediated PKA activity	↓ ↓	Wistar rat	Male	STZ induced T1DM	6-week
Fu et al., 2017	Gi expression p-TnI/TnI p-PLN^Ser16^/PLN Receptor mediated p-PLN^Ser16^/PLN	n.c. n.c. n.c. ↓	C57BL/6 mice	Male	HFD induced T2DM	8-week
Hilsted et al., 1987	Plasma cAMP level	n.c.	Human	Male	Insulin dependent juvenile onset diabetes	12–16 years
Huisamen et al., 2001	Basal AC activity G protein catalytic subunit dependent AC activity Receptor mediated AC activity Basal AC activity G protein catalytic subunit dependent AC activity Receptor mediated AC activity Basal AC activity G protein catalytic subunit dependent AC activity Receptor mediated AC activity Basal cAMP level Receptor mediated cAMP level Basal cAMP level Receptor mediated cAMP level Basal cAMP level Receptor mediated cAMP level	n.c. (6-week) n.c. (6-week) n.c. (6-week) n.c. (10-week) n.c. (10-week) n.c. (10-week) n.c. (20-week) n.c. (20-week) n.c. (20-week) ↑ (6-week) n.c. (6-week) n.c. (10-week) n.c. (10-week) ↓ (20-week) n.c. (20-week)	Zucker obese rat	na	Insulin resistant diabetes	6-, 10- and 20-week
Jiang et al., 2015	G protein catalytic subunit dependent AC activity Stimulated PKA activity	↓ ↓	Zucker obese diabetic rat	Male	Genetic T2DM	15-week
Kleindienst et al., 2016	eNOS expression p-eNOS^Ser1177^ expression p-eNOS^Thr495^expression nNOS expression iNOS expression	↑ ↑ ↑ ↓ ↑	C57BL/6 mice	Male	HF/HS diet induced T2DM	12-week
Schaffer et al., 1991	Receptor mediated AC activity Receptor mediated cAMP level Na+-K+ ATPase activity	n.c. n.c. ↓	Neonatal Wistar rat	Male	Non insulin dependent diabetes	10- and 12-month
Song et al., 2008	iNOS expression iNOS actvity	↑ ↑	ZDF rat	Male	Genetic T2DM	20-week
Wang et al., 2017	Basal cAMP level Gi expression p-PLN/PLN p-AKT expression p-TnI/TnI	↓ n.c. ↓ ↑ ↓	C57BL/6J mice	Male	HFD induced T2DM	6-month
West et al., 2019	Basal cAMP level Receptor mediated cAMP level Basal cGMP level Receptor mediated cGMP level Basal PKA activity Receptor mediated PKA activity p23–24/Total TnI p16/Total PLN	n.c. ↓ n.c. ↓ n.c. ↓ ↓ ↓	C57BL/6J mice	Male	HFD induced T2DM	4.5–5 months

AC: adenylyl cyclase, AKT: protein kinase B, cAMP: cyclic adenosine monophosphate, cGMP: cyclic guanosine monophosphate, cNOS: constitutive nitric oxide synthase, eNOS: endothelial nitric oxide synthase, Gi: inhibitory G protein, Gs: stimulatory G protein, HF: high fat, HFD: high fat diet, HS: high sucrose, iNOS: inducible nitric oxide synthase, NO: nitric oxide, NOS: nitric oxide synthase, nNOS: neuronal nitric oxide synthase, PKA: protein kinase A, PLN: phospholamban, p-AKT: phospho-AKT, p-eNOS: phospho- endothelial nitric oxide synthase, p-PKA: phospho- protein kinase A, p-PLN: phospho- phospholamban, p-RyR: phospho- ryanodine receptor, p-TnI: phospho-troponin I, RyR: ryanodine receptor, SD: Sprague Dawley, SHR: spontaneously hypertensive, STZ: streptozotocin, T1DM: type 1 diabetes mellitus, T2DM: type 2 diabetes mellitus, TnI: troponin I, WKY: Wistar Kyoto, ZDF: Zucker diabetic fatty. n.c.: no change, n/a: no data available, ↓: decreased, ↑: increased.

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
