# Peer review of "Expression and Signaling of β-Adrenoceptor Subtypes in the Diabetic Heart"

_cells, 2020, doi:10.3390/cells9122548_

Round 1

Reviewer 1 Report

I reviewed with great interest the article “Expression and signaling of β-adrenoceptor subtypes in the diabetic heart” by Betul R. Erdogan et al.

In this article, the Authors report a review regarding the β-AR expression, signaling and cardiac function mediated by β-AR in diabetic patients.

This article has exhaustively reported data from the literature, which show conflicting and non-definitive data regarding the expression of the various receptors, mainly β1- and β2-AR.

On the contrary, the literature seems to be more consistent in the demonstration of the up-regulation of β3-AR.

The work is well written, detailed, the literature up to date.

Major point

The only criticism is the poor interpretation of the results.

For example, regarding the up-regulation of β3-AR, it is possible to hypothesize a relationship between the increased expression of β3-AR and the increasing incidence of ischemia (Cheng HJ. Circ Res. 2001;89:599-606; Moniotte S. Circulation. 2001;103:1649-55)

The authors could discuss this hypothesis.

Minor points

Some typographical errors:

“because of lack of β3-AR lacks site for PKA” (line 70)

“[3H-DHA” (line 131)

Author Response

Major point

The only criticism is the poor interpretation of the results. For example, regarding the up-regulation of β3-AR, it is possible to hypothesize a relationship between the increased expression of β3-AR and the increasing incidence of ischemia (Cheng HJ. Circ Res. 2001;89:599-606; Moniotte S. Circulation. 2001;103:1649-55). The authors could discuss this hypothesis.

Reply: We appreciate reviewer’s point. While up-regulation of cardiac β3-AR in diabetes has been shown in several studies in the STZ model of T1DM, models of T2DM reported an increased (reference 101), unchanged (reference 99) or even a reduced expression (references 98 and 100) (unfortunately, the original submission mistakenly stated that reference 98 showed upregulation, when this should have said downregulation). Therefore, we have now added such hypothesis for a protective role to the Discussion (l. 474-487), but also added the caveat that this may be limited to T1DM.

Minor points

Some typographical errors:

“because of lack of β3-AR lacks site for PKA” (line 70)

“[3H-DHA” (line 131)

Reply: Thank you for picking this up. The typographical errors were corrected

Reviewer 2 Report

The paper of Erdogan et al., entitled ”Expression and signaling of beta-adrenoceptor subtypes in the diabetic heart” is a very comprehensive review regarding this subject, that is very useful for researchers involved in the study of diabetic complications”. The authors investigated more than 180 bibliographic sources in order to write this paper and have analyzed them deeply.

I reccomand to be accepted in the present form after minor spell corrections will be done.

Author Response

I reccomand to be accepted in the present form after minor spell corrections will be done.

Reply:

We thank the reviewer for his/her complimentary comments. We have carefully checked for any wording needing spelling corrections and resolved those issues.

Reviewer 3 Report

The review is interesting, once the expression and signalling pathways of β-AR in the diabetic heart may help the clinical doctors to prevent some cardiovascular events in diabetic patients. General comments:

1 - Additional human studies are necessary, since it says that there are some, but in reality the article only presents studies in rats.

2 – The section AR mediated signaling pathways, does not clearly address the importance of the various types of ion channels (mainly calcium and potassium) and the changes already observed between normal and diabetic rats.

3 - Separate the effects at the level of the ventricular muscle and the nodules, since the mechanism are different.

Author Response

1 - Additional human studies are necessary, since it says that there are some, but in reality the article only presents studies in rats.

Reply: As explained in the manuscript (l. 22-23 and 85-88), we have systematically searched PubMed using 2 different keyword combination as a basis for our manuscript. Based on the search results, we have examined more than 850 articles from published 1977 until today and we determined the related articles. As prespecified in our study plan for this review, we have referred all the studies for each species which we have obtained from this extensive search. Thus, if our manuscript discusses more studies on animals than on human, this is not a choice we made but simply reflects the existing literature. Of note, you may have overlooked that our manuscript cites 9 studies based on human tissues (references 94, 103, 113, 114, 171, 172, 173, 174, 179). However, we fully agree that the overall evidence is heavily based on animal studies which may or may not be predictive for humans, thereby creating a need for more human studies – particularly in light of the overall inconclusive findings. Additional wording in this regard has been added to the 1st paragraph of the revised Discussion (l. 456-459).

2 – The section AR mediated signaling pathways, does not clearly address the importance of the various types of ion channels (mainly calcium and potassium) and the changes already observed between normal and diabetic rats.

Reply: We agree that our manuscript did not discuss changes in cardiac ion channels and their regulation by beta-adrenoceptors in diabetes. The simple reason is that our systematic search did not yield any such studies. The apparent reason is that conclusions on such regulation probably has to be derived indirectly by first systematically searching for beta-adrenoceptor coupling to ion channels in the diabetic heart, then for changes of ion channel function in the diabetic heart. This would require an entirely different search strategy than applied for our manuscript. Within the time frame of 10 days allotted by the editor for revision, this simply was impossible. Therefore, we have added a general sentence that cardiac beta-adrenoceptors act in part by modulating ion channel activity, but that this is not discussed here in detail because our search strategy did not yield such studies (l. 265-267 and 459-461). If you feel that this is inadequate we ask that you please provide us with specific suggestions for additional references; we will be happy to incorporate them at short notice.

3 - Separate the effects at the level of the ventricular muscle and the nodules, since the mechanism are different.

Reply: We had already discussed differences within the heart in several places (e.g. l. 146-147, 193-195, and Table 1). We agree that it is likely that such differences also exist between for instance ventricular muscle and the AV node. However, our systematic search found little evidence in this regard. Thus, only reference 44 reports on AV node – and found no difference as compared to the interventricular septum. The lack of such data is now acknowledged as a limitation of the existing literature.Therefore, we cannot discuss them. If you feel that this is inadequate we ask that you please provide us with specific suggestions for additional references; we will be happy to incorporate them at short notice.

Round 2

Reviewer 3 Report

The authors have revised the manuscript according to my comments and suggestions.